Upward nitrate transport by phytoplankton in oceanic waters: balancing nutrient budgets in oligotrophic seas

Villareal Tracy A. 1 tracyv@austin.utexas.edu
Pilskaln Cynthia H. 2
Montoya Joseph P. 3
Dennett Mark 4
1 Department of Marine Science and Marine Science Institute, The University of Texas , Austin, Port Aransas, Austin, TX , USA
2 School for Marine Science and Technology (SMAST), University of Massachusetts Dartmouth , New Bedford, MA , USA
3 School of Biology, Georgia Institute of Technology , Atlanta, GA , USA
4 Woods Hole Oceanographic Institution , Woods Hole, MA , USA
Thompson Fabiano
Electronic publication date: 2014 Mar 13
Publication date: 2014
Volume: 2
Electronic Location ID: e302
Received 2014 Jan 3; Accepted 2014 Feb 12
Copyright: © 2014 Villareal et al.
Copyright year: 2014
Copyright holder: Villareal et al.
License: This is an open access article distributed under the terms of the Creative Commons Attribution License, which permits unrestricted use, distribution, and reproduction in any medium, provided the original author and source are credited.
License URL: https://creativecommons.org/licenses/by/3.0/

Keywords: Vertical migration, Diatoms, Marine, Nitrogen, Dinoflagellates, Rhizosolenia, Nitrate, Biological pump, Gyres, Mixed layer

Funding: National Science Foundation OCE-0726726 OCE-0094591 OCE-9414372 OCE-9100888 OCE-9415923 and OCE-9423471 This work has been funded by the National Science Foundation: OCE-0726726, OCE-0094591, OCE-9414372, OCE-9100888 and OCE-9415923 to TAV, and OCE-9423471 to CHP. The funders had no role in study design, data collection and analysis, decision to publish, or preparation of the manuscript.

==============================
In oceanic subtropical gyres, primary producers are numerically dominated by small (1–5 µm diameter) pro- and eukaryotic cells that primarily utilize recycled nutrients produced by rapid grazing turnover in a highly efficient microbial loop. Continuous losses of nitrogen (N) to depth by sinking, either as single cells, aggregates or fecal pellets, are balanced by both nitrate inputs at the base of the euphotic zone and N2-fixation. This input of new N to balance export losses (the biological pump) is a fundamental aspect of N cycling and central to understanding carbon fluxes in the ocean. In the Pacific Ocean, detailed N budgets at the time-series station HOT require upward transport of nitrate from the nutricline (80–100 m) into the surface layer (∼0–40 m) to balance productivity and export needs. However, concentration gradients are negligible and cannot support the fluxes. Physical processes can inject nitrate into the base of the euphotic zone, but the mechanisms for transporting this nitrate into the surface layer across many 10s of m in highly stratified systems are unknown. In these seas, vertical migration by the very largest (102–103 µm diameter) phytoplankton is common as a survival strategy to obtain N from sub-euphotic zone depths. This vertical migration is driven by buoyancy changes rather than by flagellated movement and can provide upward N transport as nitrate (mM concentrations) in the cells. However, the contribution of vertical migration to nitrate transport has been difficult to quantify over the required basin scales. In this study, we use towed optical systems and isotopic tracers to show that migrating diatom (Rhizosolenia) mats are widespread in the N. Pacific Ocean from 140°W to 175°E and together with other migrating phytoplankton (Ethmodiscus, Halosphaera, Pyrocystis, and solitary Rhizosolenia) can mediate time-averaged transport of N (235 µmol N m-2 d-1) equivalent to eddy nitrate injections (242 µmol NO3− m-2 d-1). This upward biotic transport can close N budgets in the upper 250 m of the central Pacific Ocean and together with diazotrophy creates a surface zone where biological nutrient inputs rather than physical processes dominate the new N flux. In addition to these numerically rare large migrators, there is evidence in the literature of ascending behavior in small phytoplankton that could contribute to upward flux as well. Although passive downward movement has dominated models of phytoplankton flux, there is now sufficient evidence to require a rethinking of this paradigm. Quantifying these fluxes is a challenge for the future and requires a reexamination of individual phytoplankton sinking rates as well as methods for capturing and enumerating ascending phytoplankton in the sea.

Introduction

Nitrogen (N) in the euphotic zone of the open sea has long been recognized to partition into two distinct pools of availability (Dugdale & Goering, 1967). New N represents introduction of N from outside the euphotic zone, either in the form of deep NO3−, N2-fixation, or atmospheric deposition, while regenerated N results from consumption and remineralization of dissolved or particulate N (Dugdale & Goering, 1967). While regenerated N dominates the total phytoplankton uptake, new N is critical to balance losses due to vertical fluxes and is linked to total system productivity (Eppley & Peterson, 1979). This has been expressed as the f ratio where ‘f’ = new/total N uptake and ranges from 0 to 1. On longer time scales, new N input must balance sedimentary losses or the system will experience net losses of N (Eppley & Peterson, 1979). The surface waters of the oligotrophic open ocean are considered low ‘f’ ratio environments: N and phosphate (P) often occur at nanomolar (nM) concentrations and ammonium is the dominant form taken up by phytoplankton (Lipschultz, Zafiriou & Ball, 1996; Raimbault, Garcia & Cerutti, 2008; Wu et al., 2000). The f ratio increases in the light-limited lower depths of the euphotic zone due to the increased availability of nitrate at the nutricline, thus creating what has been recognized as a two-layered structure in the Sargasso Sea of near zero f ratios in the mixed layer and elevated f ratios at or near the nutricline (Goldman, 1988). This general pattern can be modified in regions of low iron input, where iron availability limits macronutrient consumption creating regions of high nutrient-low chlorophyll (HNLC) where low phytoplankton biomass persists despite elevated nutrient concentrations (de Baar et al., 2005). These HNLC zones tend to be in equatorial or high latitude regions (Boyd et al., 2007), leaving much of the central gyres in a macronutrient (N or P) limited state. Further complexity is introduced by euphotic zone nitrification. This process introduces nitrate internally rather than from exogenous sources (Ward, 2008), can support the sustained nM nitrate concentrations ubiquitous in the gyres (Lipschultz et al., 2002) and substantially affects f-ratio calculation based on experimental 15NO3− uptake (Clark, Rees & Joint, 2008). However, it is unclear whether it can provide the produce oxygen anomalies used as geochemical signatures (Jenkins & Goldman, 1985) to calculate export loss-based new production estimates.

The nutritionally-dilute environment creates strong evolutionary pressures on phytoplankton to decrease cell size (increased surface:volume ratios) as well as for mixotrophy to supplement photosynthesis (Beardall et al., 2009; Burkholder, Glibert & Skelton, 2008). In these strongly stratified environments, small prokaryotes are numerically dominant and often are specialists for exploiting either the light-rich, but nutrient limited, upper euphotic zone, or the region at the base of the euphotic zone where light becomes limiting and nutrients increase to micromolar concentrations (Malmstrom et al., 2010). In the Pacific Ocean, this transition zone is associated with the boundary between shallow and deep phytoplankton communities of diatoms, dinoflagellates and other phytoplankton resolved by light microscopy (Venrick, 1988; Venrick, 1990). Within the phytoplankton community is also a rare, but ubiquitous, flora of giant phytoplankton (102–103 µm diameter) that avoids competition with the smaller phytoplankton by utilizing a vertical migration strategy (Villareal, Altabet & Culver-Rymsza, 1993; Villareal & Lipschultz, 1995; Villareal et al., 1999b). Buoyancy regulation rather than flagellated motility allows these taxa to migrate 50–100 + m on a multiple-day time scale, acquire nitrate in sub-euphotic zone nitrate pools, and then return to the surface for photosynthesis (Villareal & Lipschultz, 1995; Villareal et al., 1996; Woods & Villareal, 2008). Such use of sub-nutricline derived nitrate to support carbon fixation at the surface defines the process as new production and injects phytoplankton behavior into discussions of nutrient biogeochemistry.

This group of oceanic phytoplankton has unique characteristics that identify them as vertical migrators. Rhizosolenia mats, the best-studied migrators, are associations of multiple species of the diatom genus Rhizosolenia that form intertwined aggregates (Fig. 1) from <1–30 cm in size (Villareal & Carpenter, 1989; Villareal et al., 1996) and can account for 26% of particulate Si formation in the N. Pacific (Shipe et al., 1999). First observed as “confervae” by Darwin (1860) in the South Atlantic, they occur in the N. Atlantic, N. Pacific and Indian Oceans (Villareal & Carpenter, 1989). The high biomass available in single Rhizosolenia mats has made them useful general models of vertical migration in non-flagellated phytoplankton with the caveat that almost all the physiological and compositional field data are from a limited region of the eastern central N. Pacific gyre. Initially described as possessing diazotrophic symbionts (Martinez et al., 1983), subsequent work found no evidence of diazotrophy (Villareal & Carpenter, 1989). Rhizosolenia mats possess mM internal NO3− pools (INP, Villareal et al., 1996), utilize NO3− via nitrate reductase (Joseph, Villareal & Lipschultz, 1997), take up NO3−in the dark (Richardson et al., 1996), have a δ15N (3–4 per mil) similar to the deep NO3− pool (Villareal, Altabet & Culver-Rymsza, 1993), ascend at up to 6.4 m h-1, become negatively buoyant under nutrient-depletion (Villareal et al., 1996), positively buoyant as they take up NO3− (Richardson et al., 1996), and are documented down to several hundred meters by direct ROV observations (Pilskaln et al., 2005). These characteristics indicate a life cycle vertical migration to deep nitrate pools similar to the non-motile dinoflagellate Pyrocystis (Ballek & Swift, 1986), a migration notable for the greater distance (∼100 m) than that found in numerous flagellated taxa that migrate in the coastal zone (Kamykowski, Milligan & Reed, 1978). Mat consumption by the vertically migrating lantern fish Ceratoscopelus warmingii (Robison, 1984) provides at least one pathway for this C to be sequestered in the deep sea although the fate of these diatom mats is perhaps the least understood aspect of their biology. Free-living Rhizosolenia and Ethmodiscus spp, the dinoflagellate Pyrocystis spp., and the prasinophyte Halosphaera spp. each possess some subset of characteristics such as (INP) nitrate pools and buoyancy control that suggest a similar life-history characteristic (Villareal & Lipschultz, 1995). Phytoplankton migrators are clearly transporting N (and presumably P) upward, but the significance of the process in oceanic nutrient budgets has been hard to assess due to the limited geographic range of observations and abundance estimates (Emerson & Hayward, 1995; Johnson, Riser & Karl, 2010). This flora is endemic to all warm oceans, but their large size and relatively low numbers (∼100–102 m-3) have made quantification uncommon as research efforts focused on the dominant nano and picoplankton that are 6–7 orders of magnitude more abundant.

Figure 1 Rhizosolenia mats.

All scale bars are approximate. (A) Orientation view of Rhizosolenia mats in-situ. Numerous mats are evident. Station 13, 5 July 2002, 30.44°N 145.45°W (B) Individual Rhizosolenia mat. Station 13, 5 July 2002, 30.43°N 145.45°W (C) micrograph of individual mat Rhizosolenia chains. Brown regions are the nuclear mass and protoplasm of individual chains. Some cell lysis is evident due to the pressure of the cover slip. Sta. 13, 7 Sept. 1992 31.38°N 149.89°W.

Recent observations of isotopic anomalies in phytoplankton groups (Fawcett et al., 2011) and questions in nutrient budgets (Ascani et al., 2013; Johnson, Riser & Karl, 2010) have focused attention on phytoplankton sinking and ascent, and the role this may be playing in connecting deep nutrient pools with surface productivity. Nutrient budgets are key to constraining the “biological pump”, the active removal of CO2 from the surface ocean to the deep sea by biological processes (DeVries, Primeau & Deutsch, 2012). At a first approximation, use of upwelled nitrate leads to little net export of carbon (Lomas et al., 2013) since carbon dioxide is transported upward along with deep nitrate as it upwells due to advection or turbulence (Eppley & Peterson, 1979). This occurs as a result of the stoichiometric remineralization of organic material below the euphotic zone that releases CO2 proportional to the amount incorporated into the organic material at the surface. Both N and P, in general, remineralize faster than carbon (C) and decouple the stoichiometery of remineralization with depth. Remineralized C as CO2 is returned, in general, by the same processes that return nitrate to the euphotic zone. However, vertical migration and N transport by phytoplankton uncouples N and C transport. Unlike NO3− injection by physical mixing, there is no stoichiometric transport of DIC (dissolved inorganic carbon) associated with migrating phytoplankton; thus, this N use drives net drawdown of atmospheric CO2− from the euphotic zone. However, the importance of potential CO2 removal is dependent on unanswered questions surrounding the fate of these phytoplankton. In an analogous fashion, N2-fixation can support net carbon drawdown to depth since the N source (N2 gas) is uncoupled from the deep CO2 pool (Eppley & Peterson, 1979). These general relationships are key elements in the biologically-mediated summertime drawdown of dissolved inorganic carbon in the oligotrophic gyres of both the N. Atlantic and N. Pacific (Keeling, Brix & Gruber, 2004; Michaels et al., 1994).

Nitrogen budgets of the upper water column that quantify NO3− and N2-fixation inputs are therefore central to understanding the biogeochemical cycles of carbon in the euphotic zone and the remineralization region immediately below (often termed the twilight zone). At the long-term Hawaiʻi Ocean Time-series (HOT) station, annual nutrient budgets balance in the upper 250 m indicating that the NO3− component of the new N flux to support primary production and DIC drawdown is met by nitrate remineralized from sinking material in the upper 250 m. NO3− profiling technology coupled with long-term deployments of floats has highlighted the role that mesoscale eddies play in supplying NO3− to the base of the euphotic zone (∼80–100 m) (Ascani et al., 2013; Johnson, Riser & Karl, 2010; McGillicuddy et al., 2007; McGillicuddy & Robinson, 1997). However, NO3−concentrations rapidly decrease to nM levels immediately above the nutricline (∼80–100 m) (Johnson, Riser & Karl, 2010). There is no mechanism to move NO3− along this negligible diffusion gradient into the upper water column where most community production occurs and N budgets require importation of NO3− (Johnson, Riser & Karl, 2010). However, <30 µm diameter eukaryotes cells are found with δ15N signatures of 4–5 per mil at 30–60 m in the Sargasso Sea, suggesting sub-euphotic zone NO3− is reaching these depths 40+ m above the nutricline (Fawcett et al., 2011).

Phytoplankton migrating across this gradient could provide a mechanism for transport via subsurface uptake and subsequent shallow excretion and/or remineralization (Singler & Villareal, 2005). In the eastern N. Pacific gyre, vertical migration is estimated to account for an average of 14% of new production with maximum values up to 59% (Singler & Villareal, 2005; Villareal et al., 1999b). This transport has proven difficult to quantify on larger scales due to the challenges in enumerating and sampling these populations. The taxa involved, Rhizosolenia, Pyrocystis, Halosphaera, and Ethmodiscus spp. are sufficiently rare (∼100–102 cells m-3) that large water samples or nets are required to enumerate them. Migrating diatom aggregates (Rhizosolenia mats, up to 30 cm in size) are fragile, requiring enumeration and hand-collection by SCUBA divers (Alldredge & Silver, 1982; Carpenter et al., 1977). Further complication arises from the observations that small mats (∼1 cm) dominating the Rhizosolenia mat biomass are visible only with sophisticated in-situ optical sensors that overcome both contrast problems and depth limitations for SCUBA (Villareal et al., 1999b). Moreover, the recognition that in the open ocean cells <5 µm in diameter dominate uptake and remineralization has shifted focus away from the largest size fractions towards the very smallest phytoplankton (Azam et al., 1983; Hagström et al., 1988; Karl, Bidigare & Letelier, 2001; Li et al., 2011; Malone, 1980; Maranon et al., 2001).

In this paper, we present a synthesis of both literature reports and direct observations to address the broader scope of vertical migration and nutrient transport in the open sea. For vertical migration to be relevant to oceanic N cycles, migrators must be widespread, episodically abundant at levels sufficient to support the required rates, and possess the chemical and isotopic signatures of deep nitrate pools. We present new data using in-situ optical systems complemented by isotopic and abundance data that spans much of the N. Pacific Ocean. Also presented is a synthesis that documents the widespread abundance of vertically migrating Rhizosolenia mats in the Pacific Ocean and their quantitative importance in transporting and releasing N as NO3− within the upper 250 m. We also compile published data on other migrating phytoplankton in the genera Rhizosolenia, Ethmodiscus, Halosphaera, and Pyrocystis, concluding that they constitute a ubiquitous and under-sampled aspect of nutrient cycling linked directly to the behavioral characteristics of the phytoplankton. Finally, we present literature evidence that ascending behavior in smaller phytoplankton is sufficiently widespread to require a reconsideration of the role of positive buoyancy in marine phytoplankton.

Methods and Materials

Six research cruises between 1993–2003 examined Rhizosolenia mat biology along longitudinal transects at ∼28–31°N from California to Hawaii and Hawaii to west of Midway Island (Fig. 2). In all these cruises, Rhizosolenia mats were hand-collected by SCUBA divers (0–20 m) as part of a multi-year effort to enumerate and characterize their biology. Briefly, divers collected mats in polymethylpentane plastic containers (250–500 ml volume), and returned them to the ship in a closed ice chest. Mat lysis (Martinez et al., 1983) was not observed. Mats were sorted into sinking and floating mats (Villareal et al., 1996), and then the entire mat was filtered onto pre-combusted GF/F filters followed by measurement of the concentration and isotopic composition of particulate organic N and C by continuous-flow isotope ratio mass spectrometry (CF-IRMS) (Montoya, Carpenter & Capone, 2002). Details of the analytical protocol for particulate analysis as well as standardization of these isotopic measurements can be found in Montoya et al. (1996). Each analytical batch included peptone and acetanilide standards; the standard deviation of these standards was typically 0.05 per ml. Mats collected by divers are typically >2 µmol N mat-1 (Villareal et al., 1996). Samples in this range have an analytical precision of ±0.2% (Montoya et al., 1996). Isotopic composition of NO3− was measured by CF-IRMS. NO3− was first reduced to NH4+ using Devarda’s ally, followed by diffusion and trapping of the NH3 (Sigman et al., 1997). Divers enumerated mats in the upper 20 m using a 1 m2 frame equipped with a flow meter (Singler & Villareal, 2005; Villareal et al., 1996). The diver attached to the down line at 4–6 depths in the upper 20 m and swam the frame in a circle around the down line of approximately 9 m radius. The number of mats passing through the vertically oriented frame was recorded and normalized to the volume swept clear recorded on the flow meter. Integrated abundance used a trapezoidal integration to the maximum depth sampled (∼20 m) and is reported as mats m-2. In addition, abundance data were drawn from literature sources (Alldredge & Silver, 1982; Martinez et al., 1983) extending the time frame of diver-based observations to 26 years.

Figure 2 Cruise track map of sampling locations.

Cruises RNBT17WT (Mar./April 1989), W9208C (Aug. 1992), PacMat (May/June 1993), RoMP95 (June/Aug. 1995), RoMP96 (June/Aug. 1996), RoMP2002 (June 2002), RoMP2003 (Aug./Sept. 2003), POOB2008 (July 2008). Data has partially or completely presented in: RNBT17WT (Villareal & Carpenter, 1989), W9208C & PacMat (Villareal, 1993; Villareal et al., 1996), RoMP95 & RoMP96 (Pilskaln et al., 2005; Shipe et al., 1999; Villareal et al., 1999b), RoMP2002, RoMP2003 & POOB2008 (this report).

In 2003, a towed optical system (Video Plankton Recorder: VPR) was used to quantify abundance in the upper 150 m. In this data set, we employed a recalibrated and tested VPR also used in our 1996 study (Pilskaln et al., 2005; Villareal et al., 1999b). The intersection of the strobe light volume and the camera’s field of view represented an elongate trapezoid shape with a 7 cm depth of field and an individual image volume of 0.12 l. A non-reparable malfunction of the VPR-interfaced CTD on our 2003 cruise made structural adjustments necessary in order to complete the VPR surveys which involved mounting the VPR (minus its CTD) to the CTD rosette. The fin section and camera/strobe section of the VPR were separated and remounted to the CTD rosette in order to have the camera field of view extended out (∼40 cm) from the rosette frame with an unobstructed view of the water column. Additionally the fin was positioned on the top of the rosette so that the camera view remained oriented into the flow as the CTD rosette was lowered and “towed” through the water column. This orientation minimized mats contacting and fragmenting prior to photo-documentation. Ship speed was maintained at 1 knot during CTD rosette/VPR tows in which the wire-in/out speed was maintained at 12 m min-1. Four complete round-trips (one tow–yo) of the CTD rosette/VPR package between the surface and 150 m were completed at each station with a calculated water volume of 0.5 m3 viewed per each 0–150 m leg and 4.0 m3 per station tow–yo series. To provide synching of the CTD data and the VPR imagery for post-cruise analysis, a stopwatch was zeroed when the camera and strobe were turned on prior to deployment over the side. The stopwatch time was then recorded when the CTD rosette/VPR system began the first leg of the tow–yo series between the surface and 150 m and the time was recorded at the top and bottom of each 150 m leg.

VPR video from the tow–yo series completed at 10 stations and coincident with SCUBA-survey and sampling of Rhizosolenia mats in the upper 20 m was examined post-cruise. The analogue imagery from these stations was digitized and sub-sampled every 0.2 s, which assured us that we were viewing new water volume, considering the image dimensions and the ship and wire-in/out speed. The VPR data presented is from 4 of ten 2003 stations. Significant issues with the other stations’ VPR image quality (i.e., focus and electronic interference problems) and/or video recorder failures rendered the VPR imagery from 6 of the 10 stations unreliable for mat quantification. IDL and ImageJ software were used to time-link CTD data to each image, to view the collected imagery and identify Rhizosolenia mats, and to compile mat counts. Mat identification was based on their distinctive morphology of intertwining diatom chains, forming aggregations approximately ∼1 cm in size (Villareal et al., 1996), a size class rarely observed or enumerated by divers. Based on the depth occurrence of each identified Rhizosolenia mat, we calculated the mat abundance within the depth intervals of 0–20 m, 20–50 m, 50–100 m and 100–150 m.

Results and Discussion

Abundance and depth distribution of Rhizosolenia mats

Rhizosolenia mats (Fig. 1) were observed by divers at every station sampled over the 19-year period spanning the cruises (Fig. 2). At low abundance (<0.5 mat m-3), patterns in the vertical distribution were difficult to detect. At high abundance (>0.5 mats m-3), a surface maximum was often evident (up to ∼12.5 mats m-3) with decreasing abundance at depth (Fig. 3). Mats were visible below depths the divers could access and were visible to the limit of vertical visibility (40–60 m). While mats were present at all stations in Fig. 2, abundance was quite variable and occasionally (4 of 96 stations) below detection limit of the sampling frame at each of the 4–6 depths measured (∼1 mat in 30 m3). For the 1989–2003 cruises (the 2008 cruise was snorkel only with no abundance data collected), average integrated abundance determined by divers was 4.1 ± 5.7 mats m-3 with a range of 0.03–27.5 mats m-2 excluding the 4 stations where mats were below enumeration limits (Fig. 4). These values were combined with literature reports from this area in Fig. 4, generating a 26-year summary of Rhizosolenia mat distribution and diver-estimated abundance.

Figure 3 Vertical distribution and abundance of Rhizosolenia mats observed by divers.

Abundance was estimated visually using a metered frame (Villareal et al., 1996). The 1982 data are from Alldredge & Silver (1982). The remaining data (67 stations) are from cruises summarized in Fig. 2. For purposes of plotting, a zero abundance at a depth was recorded as 0.01 mats m-3. Integrated mat abundance used actual values collected. No abundance data were available from POOB2008.

Figure 4 Rhizosolenia mat integrated abundance.

Diver-collected abundance in the upper 60 m. Data are from 6 cruises spanning 1992–2003 and literature sources (Alldredge & Silver, 1982; Martinez et al., 1983). Total number of stations, n = 96. Filled squares indicate stations where mats were observed but not quantified. The A is Sta. ALOHA of the Hawaiʻi Ocean Time-Series (HOT).

The 2003 VPR imagery (RoMP 2003) revealed an abundance of Rhizosolenia mats that were ≤1 cm in size. These small-sized mats have previously been noted and are under-counted in diver surveys (Pilskaln et al., 2005; Villareal et al., 1999b). Our observations in RoMP2003 along a transect line from 168 to 177°W found mats were distributed to at least 150 m (Fig. 5). The vertical distribution had no consistent pattern with some stations (Sta. 7) showing a surface maximum, while other stations (Sta. 5) had a maximum at depth. In all cases, abundance did not decline to zero at the deepest strata (150 m). Integrated values (Table 1) ranged from 188–17,062 mats m-2. The station with two sets of tows approximately 10 h apart (Sta. 12) showed good agreement between profiles with the two samples within 2% of the mean. When VPR and diver counts were compared, divers consistently under-estimated mat abundance. The 0–150 m integrated VPR counts were up 6–2,843 times higher than the diver-based 0–20 m integrated counts (Table 1) and a comparison of Figs. 3 and 5 reveals the much higher mat densities observed in general by the VPR. VPR-based integrated abundance varied nearly 200-fold from 80–17,062 mats m-2 with 90% below diver accessible depths and had no relationship (r2 = 0.08) to diver-based abundance in the 0–20 m range (Table 1). The 2003 counts were up to 100-fold higher than VPR-based abundance data collected 2,000 km to the east in 1996 (Villareal et al., 1999b).

Figure 5 Vertical distribution of Rhizosolenia mats observed by the video plankton recorder. Data are from Aug./Sept. 2003 (RoMP2003).

Station positions are given in the figure. Data is for all sizes of mats observed by the VPR.

Table 1 N flux across the nutricline calculated from video plankton recorder (VPR) and diver-based observations made during 2003.

Flux calculations assumed 0.19 and 2.5 µmol N mat-1 (Villareal et al., 1999b) for small and large mats, respectively, and a specific rate increase of 0.14 d-1 (Richardson et al., 1998) Diver and VPR estimates are added due to the non-overlapping nature of the abundance estimates. These estimates are supplemented with contributions to upward nitrate flux from other (non-Rhizosolenia mat) migrating phytoplankton. Doubling time reflects the time required to migrate to depth, acquire nutrients, return to the surface and divide and is based on direct measurement or best available information (citations may be found in the text section on the individual taxon).

Rhizosolenia mats	
Integrated mats (mats m−2)	Sta. 5	Sta. 6	Sta. 7	Sta. 12a	Sta. 12b	
Divers (0–20 m)	26	3	3	6	6.4	
VPR (0–20 m)	188	38	3938	38	300	
VPR (0–150 m)	2,475	188	6,562	17,062	16,612	
VPR:Diver (0–20)	7	13	1313	6	47	
N flux (µmol N m-2 d-1)						
Diver-based N flux	8.9	1	1	2	2	
VPR-based N flux	64	5	170	442	430	
TOTAL (Diver + VPR)	73	6	171	444	432	
Other migrating phytoplankton	
Taxon	Abundance
0–100 m (cells m-3)	N doubling time rate (d-1)	nmol N cell-1	N flux
µmol N d-1	
Ethmodiscus spp.	1	0.09	29	3.2	
Halosphaera spp.	200	0.1	1.5	33.2	
Pyrocystis spp.	200	0.06	0.8	17.0	
Rhizosolenia spp.	50	0.14	1.6	9.2	
TOTAL				62.5	

Nitrogen isotope values

Rhizosolenia mats in all years were collected only in the upper mixed layer (<10 m depth), a zone that routinely has only nanomolar nitrate concentrations (Brzezinski, Villareal & Lipschultz, 1998; Casciotti et al., 2008; Dore & Karl, 1996). The buoyancy status of the mats is closely tied to their N status. Villareal et al. (1996) noted that when compared to negatively buoyant mats (sinkers), positively buoyant mats (floaters) have significantly higher internal nitrate pools (∼1–2 vs 8+ mM), lower C:N ratios (7–8 vs 9–11) and higher protein:carbohydrate ratios (0.4–0.6 to 1.1–1.6). These patterns are consistent with increasing nitrogen stress concurrent with the mats becoming negatively buoyant. During 2002–2003, C:N ratios in sinking mats were significantly higher than in ascending mats across the entire longitudinal gradient (Table 2), a marker resulting from unbalanced uptake of N and C and consistently tied to a vertical migration strategy (Villareal et al., 1996).

Table 2 Compositional values of Rhizosolenia mats from 2002 to 2003.

These data span from approximately 145°W to 178°E.

Year	2002	2003	
	δ15N	δ13C	C:N	δ15N	δ13C	C:N	
Mat buoyancy							
Ascending	1.38 ± 0.6 (30)	−30.41 ± 0.45 (30)	8.0 ± 0.5 (51)	2.5 ± 0.4 (95)	−30.71 ± .30 (92)	6.9 ± 1.6 (92)	
Sinking	3.6 ± 0.8 (5)	−30.41 ± 0.22 (5)	12.3 ± 1.8 (18)	3.5 ± 0.5 (34)	−30.80 ± .80 (34)	12.3 ± 0.8 (34)	

In the 2002 and 2003 samples, mat δ15N was uniformly elevated across the basin (Figs. 6 and 7) and averaged 2.91 ± 0.28 (95% C.I., n = 181) when combined with historical data (Villareal, Altabet & Culver-Rymsza, 1993; Villareal et al., 1999b; Villareal et al., 1996). Ascending mats were significantly (2.17 ± 0.32 per mil versus 3.53 ± 0.30 per mil, p = 0.05) depleted in 15N relative to sinking mats (Fig. 6, Table 2). Mats were also enriched in 15N relative to the suspended particulate material at the surface (Fig. 7). The δ15N of the ambient NO3− pool in 2002 (RoMP2002) at 200–400 m ranged from 5.29 to 6.73 per mil and was consistent with historical observations (Fig. 7). Inclusion of additional data from Station ALOHA (Casciotti et al., 2008) highlighted the lighter isotopic values of NO3− in the nutricline expected as the result of the remineralization of diazotrophically derived N. The light δ15N of mixed layer particulates is a pattern also seen at Station ALOHA and is considered typical of the oligotrophic gyres (Dore et al., 2002; Montoya, Carpenter & Capone, 2002; Montoya, Wiebe & McCarthy, 1992). The similarity of the Rhizosolenia mat δ15N to the deep NO3−δ15N is evidence that mats are generally migrating to the 150–200 m depth range. In the mats, the isotopic signature results from luxury nitrate uptake at high (µM) NO3− concentrations at depth in the dark (Joseph & Villareal, 1998; Richardson et al., 1996) and assimilation from mM INP via nitrate/nitrite reductases in the low concentration (nM) NO3− of the upper euphotic zone. Nitrate uptake does not fractionate (Comstock, 2001; Robinson, Handley & Scrimgeour, 1998) and the internal pools initially represent the source nitrate δ15N.

Figure 6 Histogram of Rhizosolenia mat δ15N. (n = 170).

Bins are 2 per mil with the lower value included in the bin and the higher value representing the upper limit. Ascending mats were statistically lighter (2.17 ± 0.32 per mil, n = 125) than descending mats (3.53 ± 0.39 per mil, n = 45). Error bars are 95% confidence intervals. Samples were collected at regular intervals on RoMP2002 and RoMP2003 (Fig. 2).

Figure 7 Particulate δ15N and nitrate δ15N of the sampled areas in the Pacific Ocean.

Suspended particulate data (open symbols) are from the 2002 cruise, pooled from Sta. 1 (22.197°N 157.960°W), 5 (28.008°N 158.019°W), 7 (28.000°N 153.736°W) and 10 (30.504°N 149.615°W). Rhizosolenia mat δ15N is averaged (box and whiskers) over all cruises (±95% C.I.). Open symbols are suspended particulate material δ15N, large solid symbols are dissolved NO3− δ15N, small filled squares are the dissolved NO3− concentration (2003 stations). Red “X” are from Casciotti et al. (2008) at Station ALOHA. Black “X” are from the 2002 stations.

Migrators cannot acquire sufficient NO3− in the mixed layer to create large INP. While rapid surge uptake at nanomolar NO3− concentrations has been demonstrated in the diatom Pheodactylum tricornutum (Raimbault, Slawyk & Gentilhomme, 1990), INP can only accumulate when assimilation via nitrate reductases is slower than uptake. Direct measurements of the migrating giant diatom Ethmodiscus indicate NO3− uptake at ambient concentrations were several orders of magnitude too low to support the observed nitrate reductase activity (Villareal et al., 1999a). Dark periods in elevated NO3− concentrations are required for INP to build up in Rhizosolenia spp. found in mats (Joseph & Villareal, 1998). Moreover, if these cells were building INP as a result of assimilating NO3− at the ambient concentrations, there should be only minimal differences between sinking and floating mats. Such uptake is inconsistent with the multiple proxies of N-limitation that are found in sinking mats and not found in floating mats (Joseph, Villareal & Lipschultz, 1997; Villareal et al., 1996). Due to the strictly internal N assimilation in the mixed layer, there is no net change of mat δ15N at the surface due to reductase fractionations and/or NO3−uptake. However, mats in both years were actively excreting NO3− (Singler & Villareal, 2005). While we could not measure the isotopic composition of this excreted N, relatively light NO3− would preferentially leak out and result in the observed pattern of lower mat δ15N in floating (recently ascended) mats compared to sinking (depleted INP).

These data provide a picture of Rhizosolenia mat abundance across the Pacific Ocean as well as within their vertical migration range. The latitudinal distribution extends from ∼24° to ∼35°N with additional observations near Oahu, Hawaiʻi (Cowen & Holloway, 1996), the coastal California current, and equatorial Pacific (Alldredge & Silver, 1982). Mats were observed over 50° of longitude (∼1/2 the width of the Pacific Ocean) and were abundant at the western terminus of the cruise set. We found no further records in the Pacific Ocean west of this point, but the broad distribution in the Indian Ocean (Carpenter et al., 1977; Wallich, 1858; Wallich, 1860), North and South Atlantic Ocean (Caron et al., 1982; Carpenter et al., 1977; Darwin, 1860), equatorial Atlantic Ocean (Bauerfeind, 1987) and north and south Central Pacific Ocean (Alldredge & Silver, 1982) supports a reasonable expectation that their distribution extends across the entire Pacific Ocean (Villareal & Carpenter, 1989). Abundance is considerably lower in the Sargasso Sea (Carpenter et al., 1977), although they are still present. Our 2003 VPR observations confirm the earlier report that small Rhizosolenia mats dominate both numerically (Villareal et al., 1999b) and in particulate Si contribution (Shipe et al., 1999) in the N. Pacific. These small mats are virtually invisible to divers due to the low contrast of small mats, and the depth limitations imposed on blue-water SCUBA techniques (∼20 m) preclude diver enumerations at depths (Villareal et al., 1999b). We conclude that the pattern of numerically dominant small mats extending to depth is the prevailing distribution of Rhizosolenia mats and that the mats are both widespread and abundant in the Pacific Ocean.

Rhizosolenia mat δ15N values show a pattern dominated by values typical of sub-euphotic zone nitrate. Prior to this study, only a handful of values were published raising the possibility that these were not representative of larger scales. However, our current data set spans nearly 1/2 the Pacific Ocean and clearly shows high δ15N NO3− pools as an N source. Vertical migration is a consistent feature of their biology and occurs across the entire distributional range. A re-assessment of the quantitative importance of mat N transport is required and is particularly timely given the need to identify mechanisms capable of closing euphotic zone nitrate budgets (Ascani et al., 2013; Johnson, Riser & Karl, 2010). In the next section, we will consider the implications for nutrient cycling and the role of ascending motion in general in phytoplankton.

Significance to oceanic nutrient cycles

The upward nitrate flux problem derives from budgeting analysis that concludes that nitrate introduced at the base of the euphotic zone must be transported upward many 10s of meters to zones of net community production and export, and that this transport occurs along a negligible concentration gradient (Ascani et al., 2013; Johnson, Riser & Karl, 2010). In order to assess the potential role of Rhizosolenia mats in the Pacific to this process, we calculated flux rates using our new data and previously published models. Nitrogen transport rates are calculated from abundance data coupled to turnover models. Negative buoyancy increases as the mats undergo progressive N limitation and sink to depth (Villareal et al., 1996). Nitrate uptake occurs at depth and in the absence of light, leading to buoyancy reversals and ascent to the surface (Richardson et al., 1996). At the surface, the pattern repeats with some fraction of the nitrate being lost via excretion (Singler & Villareal, 2005). Protist parasitism has been noted and probably results in nitrate release as well (Caron et al., 1982; Villareal & Carpenter, 1989). The overall migration cycle is shown conceptually in Fig. 8. Note that the cell growth rates of the component diatoms can be as high as 0.6 d-1 (Moore & Villareal, 1996a). Migration-based growth rates are slower due to time spent at sub-optimal light conditions while ascending/descending and during dark uptake of NO3−.

Figure 8 Conceptual model of vertical migration in Rhizosolenia mats and other giant phytoplankton.

In this simplified representation, depth intervals are given in only general terms to allow for significant life history variations in the various taxa noted in the text. Data sources: rate measurements supporting time at depth (Richardson et al., 1996), surface photosynthetic rates (Villareal et al., 1996), NO3− assimilation (Joseph, Villareal & Lipschultz, 1997), ascent rates (Moore & Villareal, 1996a; Moore & Villareal, 1996b).

Two estimates of migration-based growth rates have been published. Villareal et al. (1996) considered a simple box model that used measured mat ascent rates to calculate transit time to and from the nutricline, nitrate uptake rates based on a large diatom, and carbon-based doubling times based on oxygen evolution measurements of Rhizosolenia mats. Richardson et al. (1998) used a 13 layer model over the upper 120 m derived from Kromkamp & Walsby (1990) where mat biomass is represented in carbon and nitrogen units. Photosynthesis was modeled by a standard photosynthetic model (Platt, Gallegos & Harrison, 1980) modified to incorporate diel changes in irradiance, depth-dependent attenuation, and temperature Q10 effects. Nitrate uptake was based on Richardson et al. (1996). Ascent and descent rates were based on changes in the carbon:nitrogen ratio derived from Villareal et al. (1996). Loss rates are unknown, and were adjusted in the model to maintain a stable migration pattern for 5 cycles. The Villareal et al. model yielded total migration growth rates of 0.19–0.28 d-1, while the Richardson et al. model produced 0.11–0.15 d-1. While both models converge on values in 0.1–0.2 d-1 range, we have used a lower, more conservative estimate (0.14 d-1) for our calculations of mat turnover.

The N transport rates calculated from the VPR abundance ranged from 6–444 µmol N m-2 d-1 (Table 1) with an average daily rate of 172 µmol m-2 d-1. However, our abundance data are not uniformly distributed across the year. Rhizosolenia mat observations are biased towards June–October due to weather constraints on diving operations. We have only limited reports from April/May (Alldredge & Silver, 1982; Villareal & Carpenter, 1989) and no quantitative estimates for the balance of the year. Therefore, we restrict our calculation to a conservative 6 month distribution window to calculate the impact of only a six month period on the annual budget (i.e., 6 month rates = the annual input via migration) based on abundance at each of our stations (range = 1.1–79.9 mmol N m-2 y-1). The upper value is directly comparable to the eddy injection N (88 mmol N m-2 y-1) calculated to balance the N budget in the upper 250 m (Johnson, Riser & Karl, 2010). These calculations suggest that N transport via Rhizosolenia mats scales on an event basis that is comparable to eddy injection of nitrate to the euphotic zone, while recognizing that upward transport is not sustained at that level. This calculation is a conservative underestimate since anecdotal observations indicate mats are present year round in the eastern Pacific (Alldredge & Silver, 1982).

Finding the proper spatial and temporal scales for comparison is a challenge. Eddy injection (a physical process) and Rhizosolenia mat dynamics (a biological process) likely operate, and are certainly recorded, on different time scales. For example, the nutrient budgets were assembled for the Hawaiʻi Ocean Time-Series region at Station ALOHA (22°45′ N), a latitude that has high trade winds much of the year that inhibit diving operations. Rhizosolenia mat data were collected several hundred kilometers to the north (∼28–30°) where wind conditions permit divers to routinely enter the water. Eddy turbulent kinetic energy and numbers of eddies in the mat collection areas are low (Chelton, Schlax & Samelson, 2011) relative to Station ALHOA. We have no site where both long-term N budgets and Rhizosolenia mat abundance are available. In addition, Rhizosolenia mats are not unique in their migration strategy, and comprehensive consideration of phytoplankton upward nitrate transport requires inclusion of other migrating phytoplankton taxa. A brief review is presented here to provide the required perspective and background to justify inclusion of these taxa in the subsequent calculations.

Other vertically migrating phytoplankton taxa: life history and abundance

The literature on other migrating, non-flagellated phytoplankton in the open sea is dispersed and the natural history of this group poorly represented in the literature of the past several decades. There are several taxa that must be represented and spanning a broad taxonomic range: Pyrocystis, Halosphaera, Ethmodiscus, and free living Rhizosolenia.

Pyrocystis species are positively buoyant warm water, non-motile dinoflagellates with a dominant cyst-like non-motile stage typically 107 µm3 (Rivkin et al., 1984). They undergo a migration to the nutricline (Rivkin et al., 1984) and have been considered members of the shade flora (Sournia, 1982). Reproduction occurs by release of a brief reproductive stage from a cyst-like vegetative form (Swift & Durbin, 1971). Bilobate reproductive stages release immature vegetative stages that swell up to near full size in ∼10 min (Swift & Durbin, 1971), become positively buoyant within 13 h and indistinguishable from the cyst-like form after 15 h (Swift, Stuart & Meunier, 1976b). Thecate, dinoflagellate stages appear as swarmers in some species (Meunier & Swift, 1977; Swift & Durbin, 1971). Buoyancy reversals in the cyst form occur when negatively-buoyant nutrient-depleted stages descending to the nutricline are resupplied with NO3− and become positively buoyant, consistent with acquiring NO3− at depth (Ballek & Swift, 1986). Non-motile stage cells take up NO3− and NH4+ at almost equal rates in the light and dark (Bhovichitra & Swift, 1977) and field-collected cells at the surface contain up to 8 mM INP (Villareal & Lipschultz, 1995). Growth rates in culture range up to 0.2 div day-1 (Bhovichitra & Swift, 1977), with doubling times of 4–14 (P. fusiformis) and 10–22 days (P. noctiluca) in field populations (Swift, Stuart & Meunier, 1976a). Abundance is reported up to 200 cells m-3 in the Atlantic Ocean (Rivkin et al., 1984; Swift, Stuart & Meunier, 1976a) and 40–50 cells m-3 in the Pacific Ocean (Sukhanova, 1973; Sukhanova & Rudyakov, 1973). Photosynthetic and light acclimation curves from field populations showed a time-averaging of the light field such that C fixation at the surface supported a near-constant doubling rate throughout the euphotic zone (Rivkin et al., 1984).

Halosphaera is a genus of positively buoyant non-motile phycomate prasinophytes noted throughout the oceans (poles to tropics) from the earliest days of oceanography (Agassiz, 1906; Schmitz, 1878; Sverdrup, Johnson & Fleming, 1942). It is listed as a member of the shade flora (Sournia, 1982). Reproduction occurs by swarmer formation with up to 50,000 flagellated swarmers released from a phycoma (Parke & den Hartog-Adams, 1965). Individual swarmers can vegetatively reproduce, and then after 14–21 days start to increase in size at 5–10 µm d-1 to reach a species-specific diameter of several hundred microns. At this time, the cytoplasm undergoes numerous divisions to form flagellated swarmers (Parke & den Hartog-Adams, 1965). Size and photosynthetic rates (3–6 ng C cell-1 h-1) are similar to Pyrocystis (Rivkin & Lessard, 1986). Growth rates are poorly known; reproduction is linked to lunar rhythms in the North Sea and adjacent waters. INP up to 100 mM have been documented (Villareal & Lipschultz, 1995), and deep populations with seasonal descent and ascent are noted in the Mediterranean Sea (Wiebe, Remsen & Vaccaro, 1974). Abundance ranges from ∼10-3 cells m-3 (Wiebe, Remsen & Vaccaro, 1974) to 340 cells L-1 (Gran, 1933). Halosphaera is representative of a number of species that reproduce by phycoma and swarmer formation, including members of the genus Pterosperma. Typical concentrations reported for the Mediterranean are 1–3 L-1; Pterosperma is reported at ∼40 cells L-1 in HNLC areas of the Pacific Ocean (Gomez et al., 2007). In the text calculation on N-transport, we have assumed an abundance of 200 cells m-3 (0.2 cells L-1) as a conservative mid-range value of the 9 order of magnitude abundance range for this group.

Ethmodiscus spp. are solitary centric diatoms and are the largest diatoms known with a diameter of >2,000 µm in the Pacific Ocean; cells are somewhat smaller in the Atlantic Ocean (Swift, 1973; Villareal & Carpenter, 1994; Villareal et al., 1999a). Internal nitrate concentrations from surface-collected samples reached 27 mM in the Sargasso Sea (Villareal & Carpenter, 1994). Cells become increasing negatively buoyant as INP are depleted with positively buoyant cells having significantly higher internal nitrate concentrations than sinking cells (Villareal & Lipschultz, 1995). Nitrate reductase activity, C doubling and Si uptake rates can support doubling times of 45–75 h in large Pacific cells (Villareal et al., 1999a); cell cycle analysis suggests division rates of 0.24–0.42 div d-1 in smaller Atlantic cells (Lin & Carpenter, 1995). Pooled cells had δ15N values of 2.56–5.09 per mil (Villareal et al., 1999a). Maximum reported abundance is 27.3 cells m-3 in equatorial waters off Chile (Belyayeva, 1972), but abundance generally ranges from 0.03–4.7 cells m-3 in the Atlantic and 0.02–6 cells m-3 in the central Pacific gyre (Belyayeva, 1968; Belyayeva, 1970; Villareal et al., 2007). In the Pacific, abundance increases westward into the open Pacific Ocean with the highest values near the equator (Belyayeva, 1970). Ascent rates reach 4.9 m h-1 (Moore & Villareal, 1996b) and like Pyrocystis and Rhizosolenia, appears to result from active ionic regulation of inorganic (Woods & Villareal, 2008) and organic compounds required for osmoregulation (Boyd & Gradmann, 2002). Living cells have been collected in downward facing sediment traps at 5400 m (Villareal, 1993) indicating living cells with positive buoyancy at great depth.

Several of the Rhizosolenia species that are found in mats also exist as free-living diatom chains. These species exhibit similar characteristics to mat-forming spp. INP are present at up to 26 mM (Moore & Villareal, 1996a). Individual species (non-aggregated) ascend at up to 6.9 m h-1, depending on species and are also listed as members of the shade flora (Sournia, 1982). Growth rates for buoyant species range from 0.37 to 0.78 div d-1 in the laboratory and up to 1.0 div d-1 in the field (Moore & Villareal, 1996a; Yoder, Ackleson & Balch, 1994). Other characteristics are similar to Rhizosolenia mats (Moore & Villareal, 1996a). Little abundance information is available. R. castracanei is reported at up to 103 cells L-1 from the Bay of Naples (Marino & Modigh, 1981) and 50 cells m-3 in Sargasso Sea warm core rings (TA Villareal and TJ Smayda, unpublished data). R. debyana reached 106 cells L-1 in the Gulf of California in bloom conditions (Garate-Lizarraga, Siqueiros-Beltrones & Maldonado-Lopez, 2003); similar abundance was likely in the equatorial Pacific “Line in the Sea” front accumulation (Yoder, Ackleson & Balch, 1994).

Significance of migrating phytoplankton to the North Pacific N budget

In this final step of the calculation, we incorporated these additional migrating taxa into the model. In order to compare the spatially limited input of a mesoscale eddy with the broader distribution patterns of phytoplankton, we combined conservative abundance data and growth rate estimates for Halosphaera, Ethmodiscus, Pyrocystis and solitary Rhizosolenia spp. (Table 1) and calculated their combined contributions to NO3−flux to be 62.5 µmol N m-2 d-1. These estimates are very generalized and can only be used to scale the fluxes. Using profiler-derived estimates of eddy NO3− injection from the Pacific Ocean (Johnson, Riser & Karl, 2010), we considered the nitrate input via eddy injection over the entire time frame of measurement (145 mmol NO3− m-2 over 600 days), and computed an average daily eddy injection rate of 242 µmol NO3− m-2 d-1. Nitrate transport of Rhizosolenia mats (2002/2003 data; 172 µmol NO3− m-2 d-1) combined with other taxa (values from Table 1) equals 235 µmol N m-2 d-1. This nearly equals the average daily eddy injection of nitrate (242 µmol NO3− m-2 d-1). Our previous VPR estimates of mats (Villareal et al., 1999b) is lower, and reduces the upward transport to 179 µmol N m-2 d-1 if we include those abundance estimates. However, within the uncertainties of both calculations, this is remarkably good agreement. On a timescale of weeks to months, migrating phytoplankton can transport sufficient N from deep euphotic zone pools to the upper euphotic zone to significantly impact nutrient budgets. Upward biological transport of nitrate is quantitatively important to the biogeochemistry of surface waters in the N. Pacific gyre. Other mechanisms may exist, but migration alone appears to be sufficient to dominate the required upward transport.

The abundance range found in the vertically migrating flora is not trivial; Halosphaera abundance records span 9 orders of magnitude and the abundance used profoundly affects the calculations. While Halosphaera may be extreme, it highlights the difficulties in enumerating a frequently rare and largely ignored component of the marine phytoplankton. Moreover, there are significant gaps in our knowledge of the biology of these taxa, their life cycles and migration timing that create uncertainties in how to apply this information.

Acquisition of imported N by other phytoplankton requires release of internal nitrate pools or remineralization by grazers. Rhizosolenia mats directly release NO3−. Using excretion rates (Singler & Villareal, 2005) for NO3− (2 cruise range: 22.8–23.7 nmol N µg chl-1 h-1) and published N:Chl ratios (1.7 µmol N: µg chl a) (Villareal et al., 1996), we calculate N-specific release of ∼1.3% h-1 or up to 31% d-1. Release rates vary with Fe status, buoyancy status and location along the E-W gradient (Singler & Villareal, 2005); however, it is clear that over time scales of days to weeks, Rhizosolenia mats (and by inference, other high nitrate cells) will release NO3−. Grazers on this size class are poorly known. Hyperiid amphipods are associated with mats, as well as parasitic dinoflagellates and ciliates (Caron et al., 1982; Villareal & Carpenter, 1989; Villareal et al., 1996). Nitrate is probably released during feeding by myctophids as well (Robison, 1984). Such release by both Rhizosolenia and other ascending, high NO3− cells provides the needed mechanism for transporting NO3− to the required depths for net community production (Johnson, Riser & Karl, 2010), balancing isotope budgets (Altabet, 1989), and providing sources for the observed difference in the δ15N of NO3− in small pro- and eukaryotes (Fawcett et al., 2011). Energy dissipation via reduction of oxidized N and subsequent release also provides additional pathways to the environment from nitrate-using cells (Lomas, Rumbley & Glibert, 2000) although the isotope systematics of the various products would need to be carefully considered for their contribution to the resultant particulate signal. However, this N would contribute to meeting N budget requirements.

Using flow-cytometer sorted populations coupled with mass spectrometry, Fawcett et al. (2011) noted uniformly low δ15N values in prokaryotic phytoplankton from the Sargasso Sea while small (<30 µm) eukaryotes showed a higher mean value with significant variability suggestive of nitrate utilization from beneath the euphotic zone. Our results suggest that some of this nitrate may be made available by migrating phytoplankton. However, this requires differential availability of nitrate to the small pro- and eukaryote populations. In general, picoplankton can respond to low (<100 nM) nitrate additions (Glover, Garside & Trees, 2007). However, the response of specific taxa in this size class may not be uniform. Synechococcus, in most cases, appears to be able to utilize both NO3− and NH4+ while Prochlorococcus initially appeared unable to utilize nitrate (Moore et al., 2002), although recent work has noted evidence growth on NO3− (Casey et al., 2007). The general patterns suggest NH4+ utilization supports Procholorococcus while Synechococcus appears to have retained the capacity to utilize both NO3−and NO2− in the presence of NH4+ (Bird & Wyman, 2003; Wyman & Bird, 2007). Exceptions exist in both groups (Fuller et al., 2003) and considerable genomic diversity in N uptake exists in the two groups (Scanlan et al., 2009). Temporal changes in populations can be expected as they adapt to seasonal patterns of nutrient availability in the water column (Bragg et al., 2010). While this example focuses on prokaryotes due to the greater body of work available on oceanic forms, the relevant point is that NO3− injections above the nutricline can reasonably be expected to be differentially assimilated by subpopulations within the phytoplankton, and that, at times, significant components of the prokaryotic phytoplankton may not have access to NO3−.

Conclusions

Upward transport by phytoplankton is a quantitatively significant mechanism for transporting nutrients to the oceanic euphotic zone across broad regions of the oligotrophic open sea. Our calculations indicate that this biological flux in the Pacific can dominate the NO3− transport into the upper euphotic zone that budgets require to support the observed DIC drawdown in the summer time. There are multiple taxa involved and all oligotrophic seas possess several of them. In these large cells, NO3− excretion is probably the inevitable consequence of the mM to nM concentration gradients across the cell surface (Ter Steege et al., 1999). Although the congruence between the required N flux for budgets and the contribution from migrating flora is surprisingly good, the deeper significance of our finding is in the combined role that biology and physics play in moving essential nutrients in both directions between deep pools and the surface.

NO3−importation by the vertically migrating flora is but one component of active material rearrangement by the biota. Zooplankton diel vertical migration transports material out of the euphotic zone for remineralization and is a significant loss to the euphotic zone (Steinberg et al., 2000; Steinberg, Goldthwait & Hansell, 2002; Steinberg et al., 2008). It can represent 10–50% of the C and N flux out of the euphotic zone (Bollens et al., 2011) and up to 82% of the P flux (Hannides et al., 2009). When combined with phytoplankton vertical migration, the picture that emerges is of biological transport, both upward and downward, superimposed on both physically driven and biologically mediated new N inputs. Nitrogen-fixation coupled with NO3− release by the vertically migrating flora creates a zone of biological nutrient sources near the surface distinct from a deeper zone dominated by physical processes. In the Pacific Ocean, surface and deep phytoplankton communities persist over 1000s of km with a separation at the ∼100 m transition from nutrient- to light-limitation (Venrick, 1982; Venrick, 1999). A pattern emerges of a hydrographically structured two (or more)-layered euphotic zone with differing phytoplankton communities and biological/physical inputs of new N (Banse, 1987; Coale & Bruland, 1987; Herbland, 1983). Turbulent diffusion and eddy injection of NO3− dominates at the base of the euphotic zone; biological processes move N towards the surface and together with N2-fixation provide the community production required to close new N nutrient budgets. Atmospheric inputs may dominate truly external inputs to the surface zone (Donaghay et al., 1991).

Ascending behavior in non-flagellated phytoplankton is not limited to giant cells in the ocean. Positive buoyancy is the result of lift (cell sap density) exceeding ballast (silicate frustule in diatoms, cell wall in others) (Woods & Villareal, 2008) and theoretical considerations have suggested that there is a minimal cell size that can support positive buoyancy (Villareal, 1988). However, there is persistent evidence of positive buoyancy in smaller (10s vs 100s µm diameter) spring bloom diatoms (Acuña et al., 2010; Jenkinson, 1986; Lännergren, 1979), Antarctic diatoms (Hardy & Gunther, 1935), deep chlorophyll maximum diatoms (Waite & Nodder, 2001) and post-auxospore diatoms (Smayda & Boleyn, 1966; Waite & Harrison, 1992). Cells as small as 200 µm3 (equivalent spherical diameter = ∼8 µm) could be capable of positive buoyancy (Waite et al., 1997). These observations are scattered, but consistent with laboratory data that in sinking rate experiments, some fraction of healthy cultures are generally positively buoyant (Bienfang, 1981). Stoke’s velocities of this size range of phytoplankton are <1–2 m d-1 (Smayda, 1970); however, aggregation and chain formation could increase the effective size and the Stoke’s velocity. There are numerous aspects of this phenomenon that are unresolved, but the core observation that ascending behavior occurs in a variety of non-flagellated phytoplankton cannot be ignored.

The data that document ascending behavior in a diversity of both small and large cells are contrary to standard concepts of passive phytoplankton settling in the ocean, but are consistent with evolutionary adaptation to a physical partitioning of light and nutrient resources (Ganf & Oliver, 1982; Smetacek, 1985). We have considered only the largest vertical migrators, but persistent reports of small, ascending phytoplankton coupled with the long-noted potential of flagellated forms to vertically migrate in the open sea (Nielsen, 1939) opens entirely new linkages between events in the deep euphotic zone (Brown et al., 2008; McGillicuddy et al., 2007) and the response of surface communities. The ascent of some fraction of the biomass is a mechanism rarely considered in models of nutrient cycling in the open sea but should not be ignored. Quantifying these upward fluxes is a challenge for existing instrumentation and will likely require new approaches.

We are deeply grateful for the able assistance of numerous officers and crews of the UNOLS research vessels that supported these operations over the years. This paper is dedicated to Professor Theodore J. Smayda in celebration of his 60 years of contributions to phytoplankton ecology.

Additional Information and Declarations

Competing Interests

Author Contributions

Data Deposition

There are no competing interests. Mark Dennett is an employee of the Woods Hole Oceanographic Institution.

Tracy A. Villareal conceived and designed the experiments, performed the experiments, analyzed the data, contributed reagents/materials/analysis tools, wrote the paper, prepared figures and/or tables, reviewed drafts of the paper.

Cynthia H. Pilskaln conceived and designed the experiments, performed the experiments, analyzed the data, contributed reagents/materials/analysis tools, wrote the paper, reviewed drafts of the paper.

Joseph P. Montoya performed the experiments, analyzed the data, contributed reagents/materials/analysis tools, wrote the paper, reviewed drafts of the paper.

Mark Dennett performed the experiments, reviewed drafts of the paper, performed the at sea operation of the VPR and necessary on-site modifications to keep it working.

The following information was supplied regarding the deposition of related data:

BCO-DMO: accession number pending.

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
