# Peer review of "Upward nitrate transport by phytoplankton in oceanic waters: balancing nutrient budgets in oligotrophic seas"

_PeerJ, doi:10.7717/peerj.302_

## Round 0.1 · original submission · Minor Revisions

Dear Authors:

Two referees have now reviewed you paper carefully. Please proceed with the correction accordindly and include in your resubmittion a point-by-point response cover letter.

Reviewer 1 ·

Basic reporting

The goal of this manuscript is to compile and summarize what is known about migration by phytoplankton (as a means of acquiring nitrate and presumably phosphate) in the oligotrophic open ocean, and then to quantify the size of this biological flux and examine its importance in the context of subtropical nutrient budgets. This is a very timely subject, and I think the paper represents an important contribution in its context as a review, as well as in its estimate of the extent to which migration can contribute to upward nutrient supply. I feel that the manuscript needs some work before it can be published, however, including more exposition of the implication(s) of the calculated flux. Nevertheless, PeerJ is an appropriate journal for this work. My recommendation is somewhere between minor and major revisions prior to acceptance.

Experimental design

Methods: there is a lack of detail in the methods section regarding the N isotope analyses, which is odd given the extent of detail given to the VPR, and also does not allow me to assess the quality of the δ15N data interpreted in the paper, which the authors rely on heavily as evidence that migration is actually happening. Without the N isotope data they have evidence that the migrators are present, but not that they are migrating. More information is required in the methods section, including the quantity of N analyzed, discussion of the blanks and standards, and an indication of the accuracy and precision of the method(s).

Data: I found it difficult to work out which of the datasets presented in this study have been published before (i.e., constitute part of the ‘review’ aspect of the paper), and which are new to this manuscript. I think it’s important that the authors make this distinction clear.

Validity of the findings

N isotope data: The authors point to their δ15N data as evidence of migration; the mats exhibit a relatively lower δ15N at depth and a higher δ15N in surface waters. However, their explanation for why this pattern emerges needs to be more detailed. Granger et al., 2004 (which the authors cite to explain their findings) speaks to the efflux of high-δ15N nitrate from nitrate assimilating cells, resulting in expression of the isotope effect of nitrate assimilation. During nitrate reduction, cells accumulate large internal concentrations of high-δ15N nitrate, which is driven to such high values because of the enzyme (nitrate reductase)-level isotope effect. If the mats in the present study are accumulating large internal nitrate concentrations (which the authors say they are), the δ15N of that nitrate is likely significantly higher than the δ15N of the cell’s biomass; however, when the cells are measured via CF-GCMS, both the internal nitrate pool and the biomass will be measured together, such that the extent to which the internal nitrate pool has been converted into biomass does not matter. In other words, a cell simply assimilating nitrate into biomass (in the shallows) that it has already taken up (at depth) will not result in a change in the δ15N of that cell. Some form of efflux of high-δ15N nitrate out of the cell is required to change its δ15N. The authors do mention the possible effect of nitrate efflux, but exactly how this works is not clearly explained. Moreover, nitrate efflux will likely be accompanied by an isotope effect, albeit small, which may actually help to raise the δ15N of the effluxing mats since 14N will be preferentially effluxed from the cells. Perhaps this is not important for these data, but it underlines that more discussion of the isotope systematics is required. The isotope data are important for evincing migration by the mats, but as the paper stands, I do not feel that the explanation of these data is either complete or convincing.
Furthermore, there are other means by which the observed trend in mat δ15N can be explained. For example, shallower in the water column, the δ15N of nitrate will be higher than at the nutricline because of isotope fractionation during nitrate assimilation by phytoplankton; thus, cells assimilating nitrate at shallower depths might be expected to be higher in δ15N than those assimilating nitrate at the base of the euphotic zone even in the case that nitrate is not detectable in the water column (since the rate of its assimilation might easily match the rate of its supply). How can the authors be sure that the variations in δ15N with depth cannot be explained as assimilation of in situ nitrate rather than migration?
The authors cite Lomas et al., 2000 as a possible reason for nitrate efflux: phytoplankton may reduce nitrate to either nitrite or ammonium in order to dissipate energy, and then efflux that nitrite and/or ammonium from their cells. I agree that there is no reason that the N form effluxed by the mats need be in the form of nitrate for it to be useful to other cells. However, because of the large isotope effect of nitrate reductase, combined with the isotope effects associated with metabolic processes such as peptide dehydrolysis, any effluxed nitrite or ammonium will be low in δ15N such that its assimilation by smaller phytoplankton (i.e., in the Fawcett et al., 2011 study that the authors cite; e.g., line 440) would render these organisms low in δ15N. I don’t think this point is crucial to the present study, but it’s worth noting that the efflux of nitrite or ammonium after nitrate reduction as a mechanism to dump electrons is inconsistent with the δ15N that has been observed for small eukaryotes. Perhaps, as the authors suggest, this implies a role for migratory behavior by these small phytoplankton. Furthermore, there is no reason to suspect that nitrite or ammonium effluxed from large migrators would not be available to prokaryotic phytoplankton, which would result in a convergence of the δ15N of prokaryotes and eukaryotes.

Implications for the DIC drawdown: the authors make a rough calculation of the quantity of nitrate supply that the migrating phytoplankton mats represent, and suggest it’s equivalent to the eddy-derived supply. However, unlike the eddy-derived supply, in the case of biological transport, the supply of N is not stoichiometrically linked to carbon (as the authors point out early in the paper). Their discussion of the biologically-derived N supply would be stronger if it included mention of the DIC issue (in the section beginning at line 406): how far does the N supply that they calculate go towards resolving the seasonal DIC drawdown in the N Pacific? And how does that number compare to the Johnson et al., 2010 study? The Atlantic is different from the Pacific in that convection occurs in the late winter and early spring in the Atlantic, eradicating stratification and supply nutrients (~50% of the annual budget) to the euphotic zone. Do these differences between the two ocean gyres matter in the context of vertical migration?

Additional comments

Minor concerns and suggestions:
Various acronyms, abbreviations, and chemical formulas used throughout the paper need to be defined about first use, after which I would encourage the authors to be consistent. For example, sometimes ‘nitrogen’ is used and other times ‘N’ (which was not defined upon first use).
Furthermore, there are a fair number of typos in the manuscript that need to be corrected.

Abstract:
Line 1: I think this should say “subtropical gyres” since the subpolar gyres are subject to different hydrography and biogeochemistry
Line 23: Should this read “close N budget” rather than “nitrate budgets”?
Line 26: “Indirect evidence” for ascending behavior in small phytoplankton? As far as I know, this phenomenon has not been observed in the environment, rather it has been inferred.

Introduction:
Line 23: for the sake of completeness, atmospheric deposition of reactive N should be mentioned.
Line 34-38: it’s unclear to me what the discussion of HNLC regions adds to the paper since there is no later mention of them.
Line 39: euphotic zone
Line 45-47: I think perhaps a citation is required.
Line 59: Should “sub-nutricline” read “sub-euphotic zone”. Presumably the authors’ meaning here is below the sunlit layer and in the waters where nutrient concentrations increase (i.e., in the nutricline)?
Line 89: The citation of Johnson et al., 2010 seems inappropriate here.
Line 124: “…budgets require” what?
Line 131: Does excretion have to happen? There is much discussion of this later in the paper, but its mention here caused me to wonder whether there is a physiological reason for excretion by the large migrators such that it always happens? How important are they as an N source for other phytoplankton (non-migrators)? I don’t expect the authors to be able to answer this necessarily, but if they think that migration and excretion are always linked, that could be made clearer here in the introduction
Line 149-150: It’s actually not necessary for the δ15N of mats to reflect the δ15N of the nitrate pool since at sufficiently high ambient nitrate concentrations the isotope effect of nitrate assimilation will be expressed, rendering the mats lower in δ15N than the nitrate they are assimilating by ~5‰. While it is striking how similar the δ15N of the mats is to the nitrate supply, this does not have to be the case.
Line 157: Again, this evidence is indirect.
Figure 1: panels should be labeled A, B, C, and a scale bar is definitely required. The latitudinal/longitudinal coordinates in the caption need degree symbols.

Methods:
Please see my comments above about the methods
There are numerous places in this section where there are no citations, but rather just superscripted numbers. Please correct.
Line 176 onwards: which cruise are these data from?
Line 194: define “tow-yo” upon first use
Figure 2: Some indication of whether the data from each of these cruises has been published before would be illuminating here. I also suggest the authors label Hawaii and California.

Results and discussion:
Lines 214-239: In general, I found this section hard to decipher. I think the authors need to more clearly articulate the difference between the types of collections (i.e., as represented in Figs. 3&4 versus Fig. 5) and the depths over which they are integrating.
Also, reference to the various cruise numbers (as shown in Fig. 2) throughout this section would be useful.
Line 218: What does “clearly visible” mean? Can this be quantified?
Line 221-225: To which depths are these results referring?
Line 231: Did not drop by 150 m?
Figure 3: I’m not sure this figure clearly shows decreasing abundance as the authors suggest. I realize the x-axis is a log scale, so I think that if the data markers were made smaller, the trends might be easier to see. Also, I’m not sure how important a decreasing trend is over the upper 20 m; indeed, the authors state that 0-20 m is one of the integration intervals, and they’re looking for migrations over a greater vertical distance than 20 m. I think a little clarification is required here.
Figure 5: Does this include mats <1 cm? How should I think about this figure in the context of figs. 3 and 4?
Line 241: Are two decimals places here significant?
Line 243-245: Further explanation of this comment is required; I do not feel that the reference to Granger et al., 2004 is sufficient. The same is true of line 253-355. What is meant by “kinetic considerations”?
Figure 7: Would benefit from a legend. Also the nitrate δ15N data might be better shown by lines connecting the data markers. Where did the 2002 nitrate δ15N numbers come from? If they were measured as part of this study, some methodological detail is needed.
Line 288 onwards: what does this mean for the summertime DIC drawdown?
Line 308-317: I found this confusing. Are the authors making an annual calculation? Or a six month calculation? Is there reason to assume migratory supply of N should be similar year round? What about during periods of increased vertical mixing, such as in the late winter/early spring?
Line 338: What does “once in a lifetime” mean?
Line 339: Please define “shade flora” upon first use
Line 402: TAV?
Line 407 onwards: perhaps this is where discussion of DIC should come in.
Line 424: What is the N required for? It seems to me there are two issues with subtropical nutrient budgets: 1) geochemical tracers imply that there is far more new/export production occurring than the N supply we know of can support. However, new/export production is a whole-euphotic zone phenomenon, such that physical mechanisms of nitrate supply are likely to play an important role in reconciling the budget (eddies?), and there is no concern about the import of DIC. By contrast, 2) in the mixed layer, there is an observed DIC drawdown that occurs in the apparent absence of N and P. In order to account for this drawdown, N needs to be supplied without DIC (the authors do talk about this). So my question is which of these budgets are they hoping to balance? (obviously they’re not mutually exclusive). A little clarity or exposition on these issues, which have long plagued the oceanographic community, is required. This ties into the two-layered euphotic zone idea.
Line 467: How representative is this of the Atlantic, and should we think of that gyre in the context of the authors’ two-layered euphotic zone?
Line 481: How representative are phytoplankton with a diameter >8 microns of the oligotrophic ocean’s community? This seems larger than the average size of the more abundance eukaryotes.
Figure 8: This is a very nice summary of the paper; I think it’s highly effective. However, I am curious as to how the authors know what the timescale of nitrate acquisition at depth (hours to days) and nitrate utilization in surface waters (days) is?

Reviewer 2 ·

Basic reporting

This is a well-written article that is easy to follow. Figures and tables are presented and described clearly. The introduction is clear and comprehensive, and the results are described clearly and discussed appropriately. My only comment would be that the abstract is rather on the long side, just over 400 words. The authors might wish to cut this down a little: the first half of the abstract is basically an introduction.

Experimental design

The experimental design seems sound. The main shortcoming is that sampling was not more evenly distributed across the year, but the authors acknowledge this and explain the practical reasons, and consequently use conservative assumptions to estimate the overall importance of their results for N fluxes. I assume that the VPR is designed such that the parcel of water photographed is largely undisturbed by any bow-waves, so that the many small mats captured on video are definitely not caused by break-up of larger mats as the instrument is towed through them?
I have two other minor points:
1) The SCUBA method could be described a little better: based on what they write and their figure, I would guess that divers counted mats at ~8 discrete depths by swimming 30 m horizontally with the quadrat held upright and counted all mats within. A little more detail would be appreciated.
2) Line 232 mentions that a single sample depth was replicated with the VPR. This confuses me a little: since the VPR is a towed instrument, surely the entire depth profile would have been repeated?

Validity of the findings

The findings seem valid to me. However, I miss some discussion on whether their VPR method might have over-estimated the small-sized mats. Is there any chance that other particles might have accidentally been confused with small mats? The authors presumably believe not, and indeed they mention in the methods that some footage was discarded because technical faults made the identification inaccurate. But it would I’d appreciate some explicit discussion of the point.
Moreover, I would like the model (Lines 292 ff) to be described in more detail, rather than just referring the reader to references from the 1990s. That would also help to understand the model better. Perhaps the authors could include a table to list the model parameters, and the values they used for them?
Related to this, the authors alternate between describing their fluxes on a per year basis (Line 311) and on a per day basis (Line 410). Furthermore, they list the range of calculated annual fluxes in Line 311, but don’t give the average used later in the calculation. All this makes it a little hard to follow, so this should be tidied up.

Additional comments

Lines 98-104: I understand that this is supposed to be just a first-order approximation, but I think the authors ought to nevertheless mention that in fact N and P are remineralized faster than C, so nutrients are already somewhat decoupled from DIC. Of course, this doesn’t detract from the fact that migrating phytoplankton decouple this further.
Line 200 refers to sampling in the upper 60 m. This sounds like physical sampling of mats to me: if so, how was the 20–60 m region sampled, since they didn’t dive below 20 m? Or do they mean video sampling?
Line 369: presumably you mean 103, not 10-3?

---

## Round 0.2 · accepted · Accept

Dear Dr .Villareal We are pleased to inform you that your manuscript is accepted.